# A Novel Mutation in the *ADAMTS10* Associated with Weil–Marchesani Syndrome with a Unique Presentation of Developed Membranes Causing Severe Stenosis of the Supra Pulmonic, Supramitral, and Subaortic Areas in the Heart

**DOI:** 10.3390/ijms24108864

**Published:** 2023-05-16

**Authors:** Aviva Levitas, Liam Aspit, Neta Lowenthal, David Shaki, Hanna Krymko, Leonel Slanovic, Ronit Yagev, Ruti Parvari

**Affiliations:** 1Department of Pediatric Cardiology, Soroka University Medical Center, Faculty of Health Sciences, Ben-Gurion University of the Negev, Beer-Sheva 84101, Israel; alevitas@bgu.ac.il (A.L.); annk@bgu.ac.il (H.K.); leonelsl@clalit.org.il (L.S.); 2The Shraga Segal Department of Microbiology, Immunology & Genetics, Faculty of Health Sciences, Ben-Gurion University of the Negev, Beer-Sheva 84105, Israel; aspit@bgu.ac.il; 3The National Institute for Biotechnology in the Negev, Ben-Gurion University of the Negev, Beer-Sheva 84105, Israel; 4Pediatric Endocrinology Unit, Soroka University Medical Center, Faculty of Health Sciences, Ben-Gurion University of the Negev, Beer-Sheva 84101, Israel; netali@clalit.org.il (N.L.); shakid@bgu.ac.il (D.S.); 5Ophthalmology Department, Soroka University Medical Center, Faculty of Health Sciences, Ben-Gurion University of the Negev, Beer-Sheva 84101, Israel; yagev@bgu.ac.il

**Keywords:** Weill–Marchesani syndrome, *ADAMTS10*, aortic stenosis, RVOT right ventricular outflow tract, supravalvular pulmonary membrane, supramitral membrane, subaortic membrane

## Abstract

Weill–Marchesani syndrome (WMS) is a rare genetic inherited disorder with autosomal recessive and dominant modes of inheritance. WMS is characterized by the association of short stature, brachydactyly, joint stiffness, eye anomalies, including microspherophakia and ectopia of the lenses, and, occasionally, heart defects. We investigated the genetic cause of a unique and novel presentation of heart-developed membranes in the supra-pulmonic, supramitral, and subaortic areas, creating stenosis that recurred after their surgical resection in four patients from one extended consanguineous family. The patients also presented ocular findings consistent with Weill–Marchesani syndrome (WMS). We used whole exome sequencing (WES) to identify the causative mutation and report it as a homozygous nucleotide change c. 232T>C causing p. Tyr78His in *ADAMTS10*. *ADAMTS10* (ADAM Metallopeptidase with Thrombospondin Type 1 Motif 10) is a member of a family of zinc-dependent extracellular matrix protease family. This is the first report of a mutation in the pro-domain of *ADAMTS10*. The novel variation replaces a highly evolutionary conserved tyrosine with histidine. This change may affect the secretion or function of ADAMTS10 in the extracellular matrix. The compromise in protease activity may thus cause the unique presentation of the developed membranes in the heart and their recurrence after surgery.

## 1. Introduction

Weill–Marchesani syndrome (WMS) is a rare genetic inherited disorder described by Weill in 1932 and further defined by Marchesani in 1939. The condition is expressed as an association of short stature, brachydactyly, joint stiffness, enlarged interphalangeal joints, and eye anomalies—including microspherophakia and ectopia of the lenses, glaucoma, and occasionally, heart defects [1,2]. The primary clinical manifestation of WMS is ocular: early glaucoma, which severely compromises vision. In WMS, the lens is spherical, frequently small, and lacks evidence of microfibrils around its equator. The cardiac anomalies present in some WMS patients are generally not life-threatening [3]. Diagnosis is based on the clinical manifestations since no biochemical test or histological marker exists [4,5]. Uncommonly for a rare genetic disorder, both autosomal recessive (AR) and autosomal dominant (AD) modes of inheritance have been described for WMS [6]. WMS can be conclusively diagnosed by the identification of homozygote pathogenic variants in *ADAMTS10*, *ADAMTS17*, and *LTPBP2* or by a heterozygous pathogenic variant in *FBN1* [1,2,7,8,9]. Genetic testing is important if the clinical presentation is inconclusive or when the phenotype is indistinguishable from other connective tissue abnormalities.

*ADAMTS10* is expressed in skin, fetal chondrocytes, and fetal and adult heart [2]. The manifestations of *ADAMTS10* gene disorders in humans suggested that they participate in microfibrils’ structural and regulatory roles in extracellular matrix (ECM) function and the development of the eyes, heart, and skeleton [9,10]. ADAMTS10 regulates fibrillin microfibril biogenesis: it binds Fbn1 with high specificity and affinity at two sites and co-localizes with Fbn1 in human tissues [6]. When added to cultured fetal bovine nuchal ligament cells or fibroblast cultures, it accelerated microfibril biogenesis and assembly [6,11]. The tissue microfibrils formed by Fbn1 are crucial structural components of the ECM. Fbn1 also regulates TGFβ signaling and the biogenesis and equilibrium of elastic fibers, which support skin, ligaments, and blood vessels. Fibrilin microfibrils are the core structures of the lenticular zonules, which are compromised in WMS, and cause lens dislocations [12]. The other genes mutated in WMS encode ECM proteins that either localize to Fbn1 microfibrils in the ECM or can be involved in forming microfibrils. The heart abnormalities that can be seen in patients include patent ductus arteriosus, pulmonary stenosis, aortic stenosis, and mitral valve prolapse [2,13,14]. Thoracic aortic aneurysms and cervical artery dissection were reported in a three-generation family with FBN1-related WMS [15,16]. A systematic electrocardiogram revealed prolonged QT in individuals with WMS [14].

Here we report for the first time a rare phenotype of developed membranes in the supra-pulmonic, supramitral, and subaortic areas, creating stenosis in these areas, which recurred after surgical resection in two pairs of sisters. The patients also presented ocular findings consistent with those described in the literature for WMS. Whole exome sequencing (WES) analysis in two affected sisters revealed a novel missense homozygote variation in the pro-protein of *ADAMTS10*, segregated as expected in the family, being homozygous only in all four affected patients. This gene was previously associated with the rare connective tissue disorder—WMS. However, pathogenic variations were not described in its pro-domain, which is cleaved upon secretion from the cells; neither was it associated with the distinct phenotype of the developed heart membranes and their recurrence after surgical resection. 

## 2. Results

### 2.1. Clinical Presentation

#### 2.1.1. Patient 1

This female patient is currently 12.7 years old. She presented at two months with failure to gain weight and a heart murmur. Echocardiography showed normal anatomy and function of the left ventricle, mild right ventricular hypertrophy, and moderate supra-pulmonic valve stenosis with narrowing and stenosis of left and right pulmonary arteries. At the age of 5 months, the worsening of the stenosis required surgery for—severe supra-pulmonic stenosis, including peripheral pulmonary arteries stenosis. Patch augmentation of the right and left pulmonary arteries, the main pulmonary artery, and Tran’s annular patch were performed. She remained below the 3rd percentile for height and weight. At four years of age, she was diagnosed with severe sub-aortic stenosis with recurred supra-pulmonic stenosis. Therefore, she underwent resection of the sub-aortic membrane with pulmonary homograft replacement.

Nevertheless, three years later, a recurrence of sub-aortic stenosis was diagnosed. She is now waiting for additional surgery. The patient was followed due to short stature (height SDS—3.5 SD in childhood), delayed bone age, branchy ductility, and thickened skin joint limitation. A growth spurt was seen at puberty (height velocity: 7.5 cm/year). The patient had microspherophakia with high myopia, astigmatism, and ectopia lentis (Table 1).

#### 2.1.2. Patient 2

This female patient is currently seven years old, the younger sister of patient 1. She was diagnosed by fetal echo at 30 weeks of gestation with a stenosis of the main pulmonary arteries. She had a birth weight of 3.1 kg. The postnatal echocardiogram confirmed a mild to moderate stenosis of the supra-pulmonic valve and the right and left pulmonary arteries. By the age of eight months, her echocardiogram revealed the progress of stenosis of the supra-pulmonic valve and a narrowing of the right and left pulmonary arteries. She had cardiac surgery incorporating a pericardial patch in the right ventricular outflow tract (RVOT), including pulmonary arteries. At the age of two years, she developed a sub-aortic membrane with moderate left ventricular outflow obstruction. She is under close observation and doing relatively well. The patient was followed due to short stature (height SDS—3.5 SD in childhood), delayed bone age, branchy ductility, and thickened skin joint limitation. The patient had high astigmatism and ectopia lentis (Table 1).

#### 2.1.3. Patient 3

This female patient is the first-degree cousin of patients 1 and 2 (see Figure 1). She is currently six years old, the younger sister of patient 4. She was evaluated at the age of 12 days due to severe respiratory failure. The echocardiography revealed a supra mitral valve ring, creating severe mitral regurgitation and moderate mitral stenosis with severe pulmonary hypertension. She was intubated and treated with furosemide, nitric oxide, and inotropic support by dopamine, epinephrine, and milrinone. At the age of four months, she underwent mitral valve replacement with carbo medics 16 mechanical valves. Eighteen months later, at 22 months, she was admitted with severe respiratory failure, and echocardiography showed a stuck valve. Following a 24 h heparin infusion, a reoperation of mitral replacement was performed using a new mechanical valve. She is under observation and will require additional surgery in the future. The patient was followed due to short stature (height SDS—3.5 SD in childhood period), delayed bone age, branchy ductility thickened skin joint limitation was noticed. The patient had accommodative esotropia with hyperopia, high astigmatism, and left-eye amblyopia (Table 1).

#### 2.1.4. Patient 4 

This female patient is currently twelve years old, the sister of patient 3. She was evaluated soon after birth due to a heart murmur, thickened skin, and hand and feet swelling. Echocardiography revealed mild supra-pulmonic valve stenosis. At two and a half years old, she developed a sub-aortic membrane that worsened during follow-up, and echocardiography revealed severe sub-aortic stenosis. At the age of ten years, the patient underwent resection of the sub-aortic membrane while the supra-pulmonic stenosis remained unaltered. She is also under observation and is doing well. The patient was followed due to short stature (height SDS—3.5 SD in childhood period), delayed bone age, branchy ductility, and thickened skin joint limitation noticed. A growth spurt was noticed at puberty (height velocity: 8 cm/year). The patient had high astigmatism and left-eye amblyopia (Table 1).

The ophthalmologic findings are described in Table 1. All four patients had high astigmatism. Two patients had microspherophakia, and after pupillary dilation, the equator of the lenses was seen. One patient had very high myopia. 

All patients had stereo-acuity, the fundi examination was within normal limits, and the intra-ocular pressure examined by the I-Care tonometer was within normal limits in all four patients (below 20 mmHg).

### 2.2. Molecular Studies

#### 2.2.1. Identification of the *ADAMTS10* Mutation

The consanguinity of the family suggests that the disorder is caused by the homozygosity of a recessive mutation. The two patient sisters, III-3 and III-4 (Figure 1A), showed a unique phenotype of stenosis of all heart arteries; thus, we assumed they shared the same mutation. WES analysis was performed on both patients’ gDNA (Figure 1A). The analysis revealed sixty shared homozygous variations with allele frequencies of less than 1% in the public databases (Appendix A) (gnomad, ExAc browser, 1000 Genomes, NHLBI ESP Frequency, and dbSNP). Forty-seven variations were ruled out, as described in Appendix A. Thirteen variations remained after the filtration, as detailed in Appendix A; the variations were prioritized based on relevant mouse models, appearance as common variants, the position of the variation in the intron, change of amino acid on the exon, and family segregation. Following the prioritization, two variations remained as the potential causative mutations (in *ADAMTS10* and *OR1M1*); we consulted the physician of the patients regarding their specific phenotype, and she reported that the patients also suffer from short stature, eye abnormalities, unusually short fingers and toes, and joint stiffness; all are symptoms of Weill–Marchesani syndrome. Since the *ADAMTS10* gene is known to cause autosomal recessive Weill–Marchesani syndrome [7], it appeared that the most likely candidate variation is at chromosome 19:8670100 (GRCh37/hg19), specifically c.232T>C, p. Tyr78His in the ADAM Metallopeptidase with Thrombospondin Type 1 Motif 10 *(ADAMTS10)* gene (Gene ID: 81794; NM_030957). The variation is predicted to be deleterious by three bioinformatic models: SIFT-damaging, POLYPHEN-probably damaging, and CADD score 25.9 (values above 20 are considered damaging).

#### 2.2.2. Verification of the Variation in the *ADAMTS10* Gene

We validated the variation in the *ADAMTS10* gene identified by WES by PCR amplification using primers flanking the mutation, followed by Sanger sequencing. Next, we characterized the segregation of the variation in all available family members. All the patients were homozygous for the variation, while the rest of the healthy individuals in the family were homozygous or heterozygous for the normal allele (Figure 1A,B). The variation was not present in our collection of Bedouin exomes from 197 individuals. This variation was also absent in a database of 77 Saudis (common variants according to a database of healthy Saudi individuals with LOF in various genes [17]). Thus, its prevalence in the Bedouin population is less than 1/548. In addition, it was not documented in any public database.

#### 2.2.3. The Effect of the Pathogenic Variation on the ADAMTS10 Protein

ADAMTS10 is a member of the extracellular matrix (ECM) protease family, a disintegrin, and metalloprotease with thrombospondin motifs (Figure 2A). The family includes nineteen ADAMTS proteases and seven ADAMTS-like proteins, constituting a superfamily of glycoproteins in the ECM or remaining cell-surface-associated after secretion. ADAMTS proteases have a highly homologous N-terminal protease domain comprising a signal peptide to target them for secretion, a pro-peptide that is typically removed by furin/PACE pro-protein convertases to activate ADAMTS proteases, the catalytic metalloproteinase domain itself, and a disintegrin-like domain. Their C-terminal ancillary domains have a core region comprising a thrombospondin type 1 repeat (TSR), a cysteine-rich module, a spacer module, and a variable ensemble of additional TSRs and other modules. The C- domain is thought to endow substrate recognition, binding, specificity, and cell-surface or ECM tethering [18]. They also undergo C-terminal processing that affects substrate specificity and ECM binding [19]. The domain requires Fbn-1 microfibrils for localization in the ECM. It regulates the assembly of microfibrils and accelerates microfibril assembly through direct interactions with fibrillin-13. The ADAMTS10 protein consists of a pro-peptide domain expected to be cleaved upon exiting the cell (marked with the black arrow-furin processing site), followed by a catalytic domain and disintegrin-like domains with thrombospondin repeats which may interact with ECM19 (Figure 2A). The patient’s variation is localized in the pro-domain of the ADAMTS10 protein and replaces a highly evolutionary conserved amino acid within a conserved region (Figure 2B).

## 3. Discussion

Here we demonstrate an association between a deleterious variation in the pro-domain of *ADAMTS10* in which no variations have been reported to cause any clinical presentation and novel unique cardiac abnormalities of WMS patients [2,14,15,16]. This association is not direct proof of the casualty of the clinical presentation, which could be further verified by experiments, such as iPSC-related modeling disease or mice *AMAMTS10* mutation knock-in experiment. Not all WMS patients have cardiac anomalies [2]. Of six patients with truncating mutations in the catalytic domain of *ADAMTS10*, two had pulmonary stenosis, and two had aortic and pulmonary stenosis with dysplastic valves and hypertrophic obstructive cardiomyopathy [2]. In their study, which identified the chromosomal location of the gene causing autosomal recessive WMS, later identified as *ADAMTS10*, Faivre et al. [13] report that two out of three patients in family 1 had mild pulmonary stenosis, and the two affected individuals of family 2 had aortic and pulmonary stenosis with dysplastic valves and hypertrophic obstructive cardiomyopathy. Seven WMS patients with mutations in *ADAMTS17* had no congenital heart anomalies, and only one out of three patients with mutations in *ADAMTS10* presented congenital heart anomalies; this patient had the same homozygous mutation as his sibling, who did not have congenital heart anomalies [10]. Two patients with different missense mutations in the signal peptide of *ADAMTS10* had no congenital heart anomalies [20,21]. Three of six WMS patients, for whom the causing mutation was not provided, had valvular thickening, mitral stenosis, and mitral valve prolapse (an abnormality of the mitral valve leaflets, or supporting chords, or both which allow the leaflet(s) to prolapse into the left atrium during the ventricular contraction). ECG abnormality presented as prolonged QTc (QTc>0.46 s) [14]. Mitral valve stenosis was also reported in two brothers with WMS; the causing mutation was not provided [22]. The cardiac clinical presentation of WMS patients caused by mutations in FBN1 includes thoracic aortic aneurysm and cervical artery dissection [15,16]. No cardiac phenotype was observed in mice models (https://www.informatics.jax.org/humanDisease.shtml, accessed on 22 March 2023). A homozygous knock-in mouse model with a stop codon mutation identified in a WMS patient recapitulated the short-stature phenotype of patients and showed developmental changes to the growth plate. Moreover, it presented abnormalities in the ciliary apparatus coupled with altered distribution of Fbn1 and fibrillin-2 (Fbn2) microfibrils. The mutant mice have increased skeletal muscle mass, and muscle cells show signs of muscle regeneration/stress with decreased BMP and increased MAPK signaling. Although two out of three patients with the mutation modeled in the mouse presented mild pulmonary stenosis, no cardiac phenotype was reported for this mouse model [23]. Individuals with *ADAMTS17* mutations appear to have less severe cardiac involvement and present predominantly with the musculoskeletal and ocular features of WMS [1,18].

The four patients described here, from an extended family, have been diagnosed with WMS based on identifying a deleterious homozygous variation in *ADAMTS10*. Their ocular findings were consistent with those described in the literature for WMS, except for the absence of glaucoma. Our patients did not present structural defects in the aortic and pulmonary valves, nor stenosis, valvular thickening, or mitral valve prolapse. In addition to the previously described classical WMS features, all four patients presented developed membranes in the supra-pulmonic, supramitral, and subaortic areas, creating stenosis in these areas, which recurred after their surgical resection. Recurrence of the stenotic areas was observed following surgical resection of the membranes. The supra valvular pulmonary membrane is a constricted band above the pulmonary valve that causes supra valvular stenosis; the supra valvular mitral ring is a ridge or membrane arising from the left atrial wall overlying the mitral valve and is sometimes attached to the mitral valve annulus, variable in thickness and extension it can range from being a thin membrane to a thick fibrous ridge, and discrete subaortic membrane which presents as a membranous or fibromuscular ring below the aortic valve, also named discrete subaortic stenosis (DSS). DSS may present as a congenital heart disease, causing an increased pressure gradient in the left ventricular outflow tract (LVOT), occurring within about 6% of children with congenital heart defects and is responsible for 8–30% of total LVOT obstructions in children and up to 20% of obstructions that require intervention. It is difficult to treat and recurs in 8–34% of patients over ten years; the recurrence has been associated with young age at both initial diagnosis and surgical intervention, smaller aortic annulus, the proximity of the obstruction to the aortic valve, and a higher preoperative peak LVOT gradient. Its causes are unknown; risk factors promoting initial occurrence include morphological LVOT abnormalities such as a sharp aortoseptal angle (AoSA), sub-physiologic aortic annulus diameter, and large aortic valve-mitral valve separation distance. Females have a 1.5 times greater risk of recurrence compared to males. An increased LVOT gradient will be associated with flow acceleration, translating into an increase in the interfacial force or wall shear stress (WSS) experienced by the LVOT and the subaortic region of the septal wall. WSS plays a major role in cardiovascular disease through the alteration of endothelial cell (EC) phenotype and loss of tissue homeostasis since the histological hallmarks of DSS lesions include inflammation, fibrosis, and myofibroblast proliferation. Although the hemodynamic etiology of DSS is now relatively well accepted, its rigorous validation is still lacking. It is assumed that DSS pathogenesis may stem from a combination of hemodynamic and genetic causes (reviewed by Masse et al., 2018 [24]). Understanding the underlying mechanisms of DSS is essential because of the risks posed by surgery and the unpredictability of recurrence following resection that justify the need. In addition to wall shear stress, mechanical stress also plays an important role in calcific aortic valve disease (CAVD), the most common valvular heart disease in the aging population [25]; however, CAVD was not reported in any WMS patients.

Supra valvular pulmonary stenosis (SVPS) is considered a rare form of pulmonary stenosis (PS) and represents both a diagnostic and therapeutic challenge [26]. Little is known about the histopathology of congenital SVPS in non-syndromic patients. It may be that failure of the percutaneous approach is related to elastic recoil properties of the supra valvular ridge [26]. SVPS is considered secondary to an elastin arteriopathy, such as observed in clinical entities consistent with Williams’ syndrome [27]. Congenital mitral valve stenosis (MS) is a heterogeneous group of lesions that can occur as an isolated defect or, more commonly, in association with other left heart obstructive defects. The etiology of congenital MS is not known, although the regulatory mechanisms for this process and the specialized endothelial cells that, during embryogenesis, form the lining on the inside of the developing heart are under intensive study [28]. Presently, there are limited therapeutic approaches to treat diseases of cardiac structures arising from the endocardium. Repair or replacement of dysfunctional heart valves remains the number one need for surgery, and more than 110,000 procedures are performed in the United States each year, costing more than $2 billion [28]. Significant risk factors for survival and long-term outcomes are age at presentation, presence and severity of pulmonary hypertension, and location of the primary obstructing lesion. Anatomic features vary, and obstructing tissue or tethering structures can be present at all levels of the valve, including supra-annular, intra-leaflet, and subvalvar. Surgical techniques aim to remove abnormal tissue causing the obstruction or impediment to adequate leaflet mobility, and improve the mobility of the sub-valve structures [29].

The mutations thus far reported to cause WMS are spread out over the ADAMTS10 entire protein with no specific mutation hot spots; most mutations are located in the domains comprising the catalytic domain, and there is a notable lack of mutations found in the ancillary domains [30]. Here we report a novel missense mutation in the human *ADAMTS10* gene. The missense mutation, c.232T>C, causes the change of a conserved amino acid through the evolution in position 78, in the pro-peptide, from tyrosine to histidine. Post-translational modification has a critical impact on ADAMTS function. ADAMTS proteases are synthesized as inactive zymogens, and cleavage of the inhibitory N-terminal pro-peptide by pro-protein convertases such as furin is necessary for their activation [3]. The change of the amino acid may affect interactions in the pro-peptide domain, thus compromising the secretion of the protein from the cell into the ECM where ADAMTS10 protein is catalytically active1 or, if secreted, its function in the ECM. Proteases have crucial roles in ocular physiology and pathophysiology [31]. These proteases include matrix metalloproteases, ADAMs, and ADAMTSs. *ADAMTS10* encodes a secreted metalloprotease, which requires fibrillin-1 microfibrils for ECM localization and regulates the assembly of microfibrils [3]. Fibrillin-1 forms beaded microfibrils of 10–12 nm diameter with a typical 50–60 nm periodicity. They are widely distributed in tissues and typically associated with elastic fibers, whose assembly they guide, but they are also found in elastin-free areas, where they function independently of elastin. Among their proposed independent roles are conferring structural integrity to tissues, cell–anchorage through Arg–Gly–Asp (RGD), and heparin-binding sites in each fibrillin. Microfibrils regulate extracellular growth factor signaling, specifically by conferring latency or regulating activation of TGFβ and sequestering BMPs in the ECM [3,18]. Microfibrils were first implicated in the formation and maintenance of elastic fibers because aortic tissue from patients with Marfan syndrome (MFS), caused by mutations in FBN1 and Fbn1 mutant mice showed aberrant elastic fibers. Tropoelastin, the monomeric precursor of elastin, directly interacts with fibrillin and microfibrils coordinats elastic fiber assembly and maturation by providing a scaffold for fibulin-5 and lysyl oxidase [18]. Studies of skin fibroblasts of a child with WMS showed normal diameter and striation of collagen fibrils but abnormal extrafibrillary space and unusually large bundles of microfilaments within the cells [2,13], which may be suggestive of an impaired ECM but it is not known how. A pathogenic missense variant in *ADAMTS17*, localized in the catalytic domain, may interfere with the catalytic activity of ADAMTS17, causing the retention of ADAMTS17 in the cell or at the cell surface and its absence in a conditioned medium; additionally, abnormalities in the deposition of fibronectin while at the same time, fibrillin-1 were observed and collagen type I were retained in the secretory pathway1. The patient with this variant did not have heart manifestations. ADAMTS10 supports epithelial cell–cell junctions, binds heparan sulfate, and supports focal adhesion formation [32]. ADAMTS10 negatively regulates ADAMTS6 expression [32].

Interestingly, mutations in *ADAMTS6*, the sister protease of *ADAMTS10*, are associated with altered ventricular conduction in the heart and cause a prolonged QRS interval observed by electrocardiography [33]. In vitro validation analysis showed that the QRS-associated variants lead to impaired ADAMTS6 secretion, and loss-of-function analysis in mice demonstrated a previously unappreciated role for ADAMTS6 in connexin 43 gap junction expression, which is essential for myocardial conduction. Mutations in *ADAMTS19*, the sister protease of *ADAMTS17*, cause non-syndromic heart valve disease (dysplasia of the aortic valve in two patients and sub-aortic stenosis in another patient; Pulmonary valve stenosis in all four patients; Mitral valve dysplasia in one patient and Tricuspid valve grade I regurgitation in one patient; bicuspid aortic valve in two patients). Homozygous *Adamts19*^KO/KO^ mice revealed progressive aortic valve disease characterized by regurgitation and/or aortic stenosis but no dysfunction in the other valves. Dysfunctional aortic valves showed thickening of commissures and reduced opening of the valve. Histological examination of dysfunctional aortic valves from *Adamts19*^KO/KO^ mice showed disorganization of the extracellular matrix (ECM) throughout the valve leaflets compared to controls. The authors hypothesized that loss of *Adamts19* perturbs shear stress signaling in the endothelial cells of the aortic valve, inducing over-expression of the transcription factor Klf2. Over time, defects in this mechano transduction pathway lead to increased cellularity and proteoglycan deposition in the valves, ECM disorganization, and heart valve disease [34]. No substrates for ADAMTS19 have been reported.

Our study is the first to demonstrate a mutation in the pro-peptide domain of ADAMTS10 in humans associated with a novel pathology in the heart. The pathology could arise from the absence of the protease from the cell surface or the ECM or from the acquisition of a dominant negative function due to the aberrant retention of the mutated ADAMTS10 protease in the secretory pathway. Patients with missense mutations in the signal peptide do not present congenital heart anomalies. They do not secrete ADAMTS10, which appears to have a reduced polypeptide concentration in the endoplasmic reticulum [20] or significantly diminished secretion [21]. This finding suggests that the underlying mechanism causing the appearance of the membranes in our patients is probably not just the absence of ADAMTS10 in the ECM. Our study may shed light on a mechanism involved in the development of the membranes in the heart. Further research is required to better understand the exact mechanism in which the activity of this metalloprotease takes action and its effect on the development of the heart membranes.

## 4. Materials and Methods

### 4.1. Clinic

The cardiac evaluation included echocardiography with Doppler and Color Doppler using the GE Vivid E95 with the 6SD and MS5 transducers. Cardiac anatomy and function were evaluated using two-dimensional (2D) and Doppler echocardiography (TTE) in the parasternal long and short axis, apical four-chamber, sub-costal, suprasternal notch, and right parasternal views. Twelve-lead electrocardiography (ECG) was performed on all the patients.

An ophthalmologic examination was performed by slit lamp. The fundi examination and the cycloplegic refraction were performed 45 min after the instillation of sterile eye drops of 1% Cyclopentolate Hydrochloride and 2.5% Phenylephrine twice, ten minutes apart. The fundi examination was done by indirect ophthalmoscopy.

### 4.2. Molecular

Genomic DNA was extracted from the patient’s blood. Whole exome sequencing (WES) analysis was carried out at Theragene. SureSelect XT V6 was used for library preparation, the target size was 58 Mb, and the coverage uniformity at ×10 coverage was ≥90%. Results were analyzed using QIAGEN’s Ingenuity Variant Analysis software (www.qiagen.com/ingenuity, accessed on 22 March 2023, QIAGEN, Redwood City, CA, USA). PCR amplification of exon 4 of the *ADAMTS10* gene (Gene ID: 81794, NM_030957) was performed using primers forward: 5′ CTGGAGAGCTATGAGATCGCCT 3′ and reverse: 5′ CCTCCACAGGTGCTGATGG 3′ (annealing temperature 64 °C). Direct sequencing of the PCR products was performed as detailed [35].

## Figures and Tables

**Figure 1 ijms-24-08864-f001:**
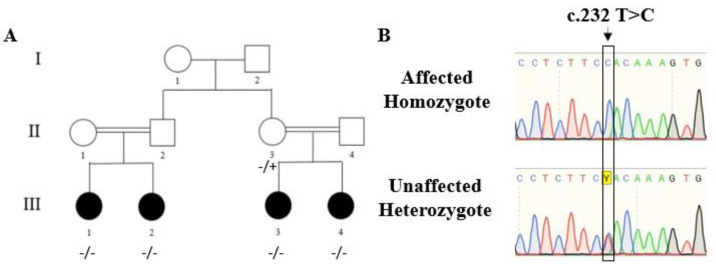
Family pedigree and identification of the pathogenic variation. (**A**) Family pedigree and segregation of the variation in the family. I, II and III—generation; 1–4—numbers of individuals in generation − variation, + reference sequence. (**B**) Sanger sequencing demonstrates homozygosity for the mutation and heterozygosity.

**Figure 2 ijms-24-08864-f002:**
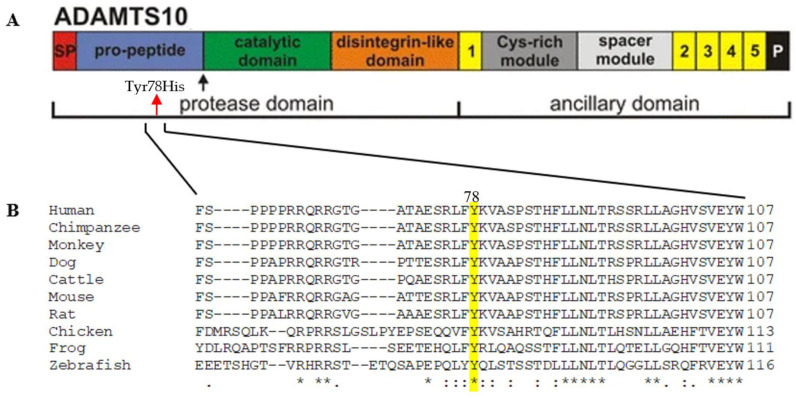
Effect of the mutation on the Adamts10 protein. (**A**) Schematic presentation of the domain structure of ADAMTS103. The arrow points to the furin cleavage site. The variation is predicted to map in the pro-peptide domain. The approximate site of the variation is marked by a red arrow. (**B**) Evolutionary conservation of the amino acid (Tyr78) changed by the variation (marked in yellow). The multiple alignment was done by ClustalOMEGA. An * (asterisk) indicates positions with a single, fully conserved residue. A : (colon) indicates conservation between groups of strongly similar properties A . (period) indicates conservation between amino acids of weakly similar properties.

**Table 1 ijms-24-08864-t001:** Cardiac and ophthalmic involvement in each patient.

Patient	Cardiac Involvement	Ophthalmic Involvement
1	1. Severe Supra-pulmonic valve stenosis RPA + LPA stenosis 2. Discrete sub-aortic membrane: Severe sub-aortic stenosis	Microspherophakia High myopia Astigmatism Ectopia lentis
		Refraction: Rt eye: −11.00–5.25 × 177, Lt eye: −10.25–2.50 × 37
		Stereopsis: 140 s of arc
		Worth 4 Dot test: fusion for near and distance. Pupillary dilatation revealed the lenses to be clear and small. The lower and temporal equator of the lenses could be seen in the dilated pupils. Anterior segments and fundi were within normal limits.
2	1. Severe supra-pulmonic valve stenosis RPA + LPA stenosis 2. Discrete sub-aortic membrane: moderate sub-aortic stenosis	Visual acuity: in both eyes 20/30 with glasses
		Refraction: Rt eye: −1.00–5.00 × 180, Lt eye: −2.00–4.50 × 20
		Stereopsis: 140 s of arc
		Worth 4 Dot test: fusion for near and distance. Pupillary dilatation revealed the lenses to be clear and small. The temporal equator of the Rt lens and the lower equator of the Lt lens could be seen in the dilated pupils. Anterior segments and fundi were within normal limits.
		High Astigmatism
3	Supramitral valvular ring, mitral stenosis: moderate stenosis and severe mitral regurgitation	Visual acuity: Rt eye 20/25, Lt eye 20/50 with glasses
		Refraction: Rt eye: +1.50–5.25 × 140, Lt eye: +2.50–5.75 × 25
		Stereopsis: 140 s of arc
		Worth 4 Dot test: fusion for near and left eye suppression for distance. The patient had microstrabismus of Lt eye 4PD for distance. Anterior segments, lenses, and fundi were within normal limits.
		High Astigmatism
4	1. Mild supra-pulmonic valve stenosis 2. Discrete sub-aortic membrane: moderate sub-aortic stenosis	Visual acuity: Rt eye 20/40, Lt eye 20/60 with glasses
		Refraction: Rt eye: +4.00–5.00 × 170, Lt eye: +5.00–5.75 × 10
		Stereopsis: Fly test positive
		Worth 4 Dot test: fusion for near
		The patient had accommodative esotropia. Anterior segments, lenses, and fundi were within normal limits.

RPA—Right pulmonary artery. LPA—Left pulmonary valve stenosis. The numbers of the patients are represented in Figure 1A in the III generation.

## Data Availability

All data are included in the manuscript.

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
