# Peer review of "A Novel Mutation in the ADAMTS10 Associated with Weil–Marchesani Syndrome with a Unique Presentation of Developed Membranes Causing Severe Stenosis of the Supra Pulmonic, Supramitral, and Subaortic Areas in the Heart"

_ijms, 2023, doi:10.3390/ijms24108864_

Round 1
Reviewer 1 Report
This manuscript reported a case study of four patients from an extended consanguineous family who presented with a unique and novel presentation of heart-developed membranes in the supra pulmonic, supramitral, and subaortic areas, creating stenosis that recurred after their surgical resection. The authors used whole exome sequencing to identify a homozygous nucleotide change in the ADAMTS10 gene as the causative mutation, and identified a correlation between a variation in the ADAMTS10 gene and unique cardiac abnormalities in some patients with Williams-Beuren syndrome (WMS). Overall the subject is interesting and worth publication.
The authors discuss the challenges of treating these abnormalities and note the role of wall shear stress in cardiovascular disease. Other than wall shear stress, the mechanical stress also plays an important role on such cardiovascular disease (see https://royalsocietypublishing.org/doi/pdf/10.1098/rsif.2019.0893), I suggest the authors to add this to discussions.
Author Response
We thank the reviewer for his thoughtful review and his suggestion to add the important reference.
We have added this reference “Other than wall shear stress, mechanical stress also plays an important role in calcific aortic valve disease (CAVD), the most common valvular heart disease in the aging population25, however, CAVD was not reported in any WMS patients. “ Page 8, lines 295-297. (Highlighted yellow).
Reviewer 2 Report
The authors studied the genetic cause of pulmonary artery stenosis in the four patients from the same family with Weill-Marchesani syndrome (WMS) and identified causative mutation using whole exome sequencing. The authors reported a homozygous nucleotide change causing protein mutation in Tyr78His in ADAMTS10, which is a member of a family of zinc-dependent extracellular matrix protease family. The novel variation replaced a highly evolutionary conserved tyrosine with histidine. This mutation may cause the dysfunction of ADAMTS10 in the extracellular matrix and may explain the pathological phenotype of WMS patients.
This is a novel study to identify potential causal mutations for rare heart disease. It is helpful for identifying a new diagnosis marker for WMS.
Major
It remains unclear whether those four patients were caused by single mutation from ADAMTS10. Further verified experiments, such as iPSC related modeling disease or mice AMAMTS10 mutation knock-in experiment, may be helpful.
Minor
Please provide conserved information across different species for amino acid Tyr78.
Author Response
We thank the reviewer for his thoughtful review and suggestions.
In reply to the major concern we have added this concern in the discussion, “This association is not direct proof of the casualty of the clinical presentation, which could be further verified experiments, such as iPSC-related modeling disease or mice AMAMTS10 mutation knock-in experiment. “ page 6 lines 230-233. (highlighted yellow). Unfortunately, the experiments suggested by the reviewer are beyond the scope of the present study.